# A Powerful Joint Modulation and STBC Identification Algorithm for Multiuser Uplink SC-FDMA Transmissions

Mohamed Marey [1,*,†] and Hala Mostafa [2,†]

1   Smart Systems Engineering Laboratory, College of Engineering, Prince Sultan University,
    Riyadh 11586, Saudi Arabia
2   Department of Information Technology, College of Computer and Information Sciences, Princess Nourah bint
    Abdulrahman University, P.O. Box 84428, Riyadh 11671, Saudi Arabia
*   Correspondence: mfmmarey@psu.edu.sa
†   These authors contributed equally to this work.

**Abstract:** Owing to the rapid development and broad adoption of multiple antenna communication systems over the past few years, space-time block coding (STBC) identification has emerged as a crucial responsibility for smart radios. The majority of previous analysis of STBC identification assumed that the utilized modulation schemes for single-user and uncoded broadcasts were known. This paper investigates the challenge of joint STBC and modulation identification for uplink transmissions with numerous users in single-carrier frequency division multiple access (SC-FDMA) systems using coded transmissions. Multi-user channel estimation brings us one step closer to implementing the proposed design in real-world systems. We additionally employ the channel decoder's deliverables, which are common in many real-world systems, to enhance the identifying and estimating procedures. Mathematical findings prove that a recursive approach can be utilized to tackle the maximum likelihood (ML) problem of simultaneous STBC and modulation identification with channel estimation. Distinguishing the superimposed signals that originate at the base-station (BS) is accomplished with the use of the space-alternating generalized expectation-maximization (SAGE) algorithm. After that, an expectation-maximization (EM) engine is deployed to make the necessary adjustments to the parameters being considered for each user. The success of the above-mentioned architecture for usage in practical applications is demonstrated by the simulation results obtained under various conditions.

**Keywords:** signal identification; multiuser transmissions; SAGE procedure

## 1. Introduction

Assessment of wireless signals, with the intention of determining the precise broadcast parameters of the employed emitter, has been a popular topic of study for decades. The term "signal identification" is commonly used to describe to this type of investigation, which has both military and civilian purposes [1–3]. Signal hacking; wireless surveillance; interference diagnosis and neutralization; and electronic warfare are just a few military applications that have made extensive use of this technology for a long time [4]. The development of smart radios has recently resulted in increased involvement of signal recognition technologies in the frame of reference of industrial uses, such as cellular mobile systems and wireless Local area networks. These radios are reprogrammed, giving them the capacity to change their broadcast configurations, including things such as modulation schemes [5–8] and channel error-correcting code types [9–12].

The likelihood-based (LB) and the feature-based (FB) strategies are two of the most used ways to identify wireless signals. The LB solutions have great identification performance because they construct a probabilistic representation of the collected signals [13,14]. However, they suffer from a high computing overhead and demand prior awareness of channel parameters. On the other hand, FB methods are designed to identify signals by first extracting their most distinctive properties, such as cyclostationary patterns and

signal statistics [15–18]. Typically, FB-based algorithms are able to reduce complexity while maintaining a satisfactory level of identification performance.

When designing an identifier for multiple-input multiple-output (MIMO) antenna systems, it is essential to take into consideration the specific technical challenges that are presented by the sort of antenna elements layout. STBC stands for space-time block coding, whereby a numerous duplicates of an input data are emitted in various time slots using many broadcast antennas. This allows for diversity to be achieved while maintaining a basic receiver topology. STBC identification methods require prior knowledge of the modulation type in order to perform properly [19] and modulation detection techniques for STBCs typically serve based on the information that they are aware of the code form [20,21]. As a consequence, it is possible that sequencing different kinds of these algorithms is not viable in practice. This demonstrates the importance of providing co-recognition algorithms, which can determine the STBC and the modulation schemes in a single operation.

### 1.1. Related Works

Researchers have recently shown a lot of enthusiasm for the challenge of identifying modulation in MIMO broadcasts. Related efforts have been accomplished for both single-carrier and multi-carrier systems. A decision theoretic algorithm for use in spatially multiplexed MIMO systems is offered for determining the number of transmit antennas and identifying modulation simultaneously [22]. Based on the correlation functions of the received signals, a binary-tree algorithm is developed to identify the type of modulation used by MIMO systems operating over unexplored wireless multipath fading channels [23]. The properties of the fourth-order cumulants of the predicted broadcast signal streams and a likelihood assessment are employed to implement a modulation identification algorithm for MIMO systems [24]. Combining the qualities of the correlation functions and cyclic cumulant results in the development of a modulation identifier used for MIMO systems over frequency-selective channels [25]. A modulation identifier is designed using Gibbs sampling and mean field variational inference for MIMO-OFDM systems operating on wireless channels with unknown characteristics [14]. A semi-supervised deep learning technique for MIMO modulation identification was developed in [26] by framing multi-class classification as an anomaly inference issue with unlabeled data. A three-dimensional convolutional neural network with cuboidal filtering was proposed in [27] for modulation identification in MIMO and orthogonal frequency division multiplexing (OFDM) transmissions. A tree-based blind modulation classification algorithm for asynchronous MIMO-OFDM systems was created in [28] by computing higher-order cumulants of the collected signal in the frequency domain. Three distribution tests were used in [29] to compare the possible modulation options, and their results were combined by a multi-layer classifier to boost the eventual classification accuracy. The authors of [30] proposed a modulation identifier for MIMO systems using a convolutional neural network with zero-forcing (ZF) equalization and investigated the effect of incomplete channel state information on identification accuracy.

In the meanwhile, a large body of literature has been devoted to STBC categorization for both single-carrier and multicarrier MIMO transmissions. A set of techniques dependent on the fourth-order moment was created in [31,32] to discriminate between Alamouti (AL) and spatial multiplexing (SM) STBC transmissions across Nakagami single-path wireless channels. The utilization of the second-order cyclostationarity of two distinct received signals was assessed in [33,34] in order to categorize among various STBC signals. The authors of [35] employed the scattering capabilities of frequency-selective wireless channels in order to differentiate between AL and SM STBC transmissions. A maximum-likelihood (ML) strategy [36] and a Frobenius norm approach [37] were evolved to make distinctions between STBC broadcasts. The researchers of [38–41] utilized the cross-correlation that exists between multiple collected signals in order to identify AL and SM broadcasts within the context of OFDM systems. Training symbols are adopted to categorize amongst STBCs for OFDM systems [42].

*1.2. Contributions*

All of the aforementioned works are geared toward either conducting modulation identification or STBC recognitions for MIMO systems. However, practical systems necessitate paired conducting modulation and STBCs identifications in order to function properly. There are just two published studies on the joint identification of the previously mentioned parameters, and both of them focus exclusively on single-user and single-carrier transmissions [43,44]. The following is a list of the notable contributions that this work delivers.

- We introduce a new strategy for jointly classifying STBC and modulation parameters for usage in multi-user broadcasts of upstream single-carrier frequency division multiple access (SC-FDMA) systems.
- The mathematical investigation that is presented in this work demonstrates that the precise ML solution to this problem is too complicated to be used in practical settings. As a result, we have created a new mechanism that operates in a recursive fashion. The space-alternating generalized expectation-maximization (SAGE) procedure [45,46] is employed to separate the layered signals that have arrived at the BS.
- The expectation phase of the SAGE algorithm takes advantage of the soft information delivered by the channel decoder in order to limit the amount of multiple access interference that is produced by other users. This results in a search strategy that is comprised of several one-dimensional scans, as opposed to the more complex multi-dimensional scan.
- The proposed method of identification is supplemented by the creation of estimates of the channel impulse responses of the existing users.

  The proposed approach has several benefits, including those listed below.

- As long as the decoding procedure is a soft-decision one [47], the suggested classification method operates with any error-correcting code. Due to this, the suggested algorithm can easily be incorporated into the currently used wireless standards, providing a great deal of flexibility.
- The developed classification algorithm is dynamic in that it can be used to a wide variety of STBCs and modulation techniques with no special adjustments needed.

The remaining portion of the work is divided up into the following parts. Section 2 explores the formulation of the challenge under consideration and the system modeling. Section 3 details the identification algorithm that is being proposed. Section 4 reports on practical concerns and interpretations. Section 5 presents and analyzes the simulation findings. Section 6 offers the closing observations.

## 2. Signal Model and Problem Formulation

We explore the upstream broadcast of a SC-FDMA system, where $U$ users are in simultaneous communication with a single base-station (BS), as illustrated in Figure 1. The different notations are explained in Table 1. Each user is allocated a portion of the total number of accessible subcarriers, $\mathcal{N}$, denoted by $\Phi(u)$, where $u$ is the user index. The $u$th user has exclusive usage of the subset $\Phi(u)$, rather than sharing it with any other users. This fulfills that $\bigcup_{u=0}^{U-1} \Phi(u) = \{0, 1, \cdots, \mathcal{N} - 1\}$ and $\Phi(u) \bigcap \Phi(u') = \phi$ for $u \neq u'$, and $\phi$ is the empty set. In this context, we impose no restrictions on the modulation, coding, or interleaving settings employed by each user.

**Table 1.** Notation.

| Notation | Meaning |
|---|---|
| $U$ | Total number of users |
| $u$ | User index |
| $\mathcal{N}$ | Total available number of subcarriers |
| $\Phi(u)$ | The subcarrier set allocated to user $u$ |
| $\langle\Phi(u)\rangle$ | The cardinality of $\Phi(u)$ |
| $\phi$ | The empty set |
| $\Omega(u)$ | The modulation type of user $u$ |
| $\alpha(u)$ | The STBC format of user $u$ |
| $\mathbf{d}_z^{(u)}$ | The $z$th data sequence of user $u$ |
| $\mathbf{F}_z^{(u)}$ | The $z$th frequency-domain SC-FDMA sequence of user $u$ |
| $\mathbf{s}_z^{(u)}$ | The $z$th time-domain SC-FDMA sequence of user $u$ |
| $\bar{\mathbf{s}}^{(u)}$ | The input to the encoder of the selected STBC format |
| $P^{(u)}$ | The number of transmit antennas of user $u$ |
| $Z_2^{(u)}$ | The number of STBC time intervals of user $u$ |
| $Z_1^{(u)}$ | The number of symbols fed into the STBC encoder of user $u$ |
| $\mathbf{h}^{(u,p)}$ | The CIR between $p$ transmit antenna of user $u$ and the base-station |
| $\mathbf{r}$ | The received signal at the base-station |
| $\odot$ | The linear convolution between two vectors |
| $\mathbf{w}$ | The noise vector at the base-station |
| $\mathbf{C}_{\alpha(u),\Omega(u)}^{(u,p)}(v_1,v_2)$ | The component situated at row $v_1$ and column $v_2$ of matrix $\mathbf{C}_{\alpha(u),\Omega(u)}^{(u,p)}$ |
| $\bar{\mathbf{c}}_{\alpha(u),\Omega(u)}^{(u,p)}(v_1-v_2)$ | The $(v_1-v_2)$th component of vector $\bar{\mathbf{c}}_{\alpha(u),\Omega(u)}^{(u,p)}$ |
| $\Pr(\circ|\nabla)$ | The probability density function of $\circ$ given $\nabla$ |

### A. Transmitters

To compensate for errors caused during transmissions, a channel encoder with rate $c(u)$ adds redundancy digits to a stream of binary data of user $u$. The produced bits are interleaved and projected to complex data symbols using a predefined signal pattern $\Omega(u)$ of unit energy. User $u$ optimizes its broadcasting modulation technique $\Omega(u)$ on time-varying running circumstances in order to increase net throughput and minimize energy usage while maintaining a predetermined level of service. A small number of training symbols are embedded into the sent information to kick off the identification mechanism, as explained later. The final series comprises of $Z(u)$ sequences, each of which contains $\langle\Phi(u)\rangle$ symbols. Here, $\langle\Phi(u)\rangle$ denotes the cardinality of $\Phi(u)$. We define $\mathbf{d}_z^{(u)} = \left[d_z^{(u)}(0),\cdots,d_z^{(u)}(\langle\Phi(u)\rangle-1)\right]$ as the $z$th sequence, where $d_z^{(u)}(m)$ is the $m$th element of the sequence. After performing a fast Fourier transform (FFT) algorithm on $\langle\Phi(u)\rangle$ points, we acquire frequency-domain observations $\mathbf{F}_z^{(u)} = \left[f_z^{(u)}(0),\cdots,f_z^{(u)}(\langle\Phi(u)\rangle)-1\right]$, which we express as

$$f_z^{(u)}(l) = \sum_{m=0}^{\langle\Phi(u)\rangle-1} d_z^{(u)}(m)\exp(-j2\pi ml/\langle\Phi(u)\rangle), \tag{1}$$

where $f_z^{(u)}(l)$ is the $l$th sample of $\mathbf{F}_z^{(u)}$ and $j = \sqrt{-1}$. The proposed approach can be employed with either frequency-interleaved or frequency-localized translation [48]. The vector $\mathbf{F}_z^{(u)}$ is turned into a time-domain SC-FDMA symbol, $\mathbf{s}_z^{(u)} = \left[s_z^{(u)}(-\nu),\cdots,s_z^{(u)}(\mathcal{N}-1)\right]$, by employing an $\mathcal{N}$-point inverse FFT algorithm with attaching a cyclic prefix of length $\nu$ as

$$s_z^{(u)}(m) = \sum_{l=0}^{\mathcal{N}-1} f_z^{(u)}(l)\exp(j2\pi ml/\mathcal{N}), \tag{2}$$

where $m = -\nu,\cdots,\mathcal{N}-1$. User $u$'s time-domain symbols, $\bar{\mathbf{s}}^{(u)} = \left[\mathbf{s}_0^{(u)},\cdots,\mathbf{s}_{Z^{(u)}-1}^{(u)}\right]$, are input into a space-time encoder, which broadcasts each $Z_1^{(u)}$ symbol through $P^{(u)}$ antennas

in $Z_2^{(u)}$ time intervals. For illustration, the STBC of parameters $Z_1^{(u)} = 2$, $Z_2^{(u)} = 2$, and $P^{(u)} = 2$, named Alamouti code [49], sends two symbols $\mathbf{s}_z^{(u)}$ and $\mathbf{s}_{z+1}^{(u)}$ via two antennas in two successive time intervals. The first antenna delivers $\mathbf{s}_z^{(u)}$ and $-\mathbf{s}_{z+1}^{(u)*}$ in two time intervals, whereas the second one sends $\mathbf{s}_{z+1}^{(u)}$ and $\mathbf{s}_z^{(u)*}$ during the same periods.

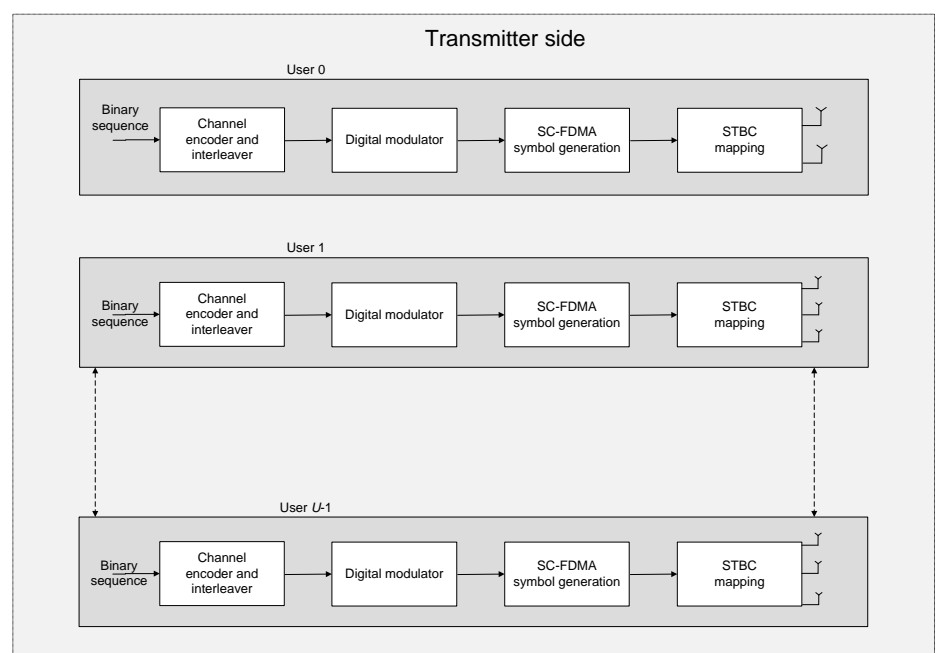

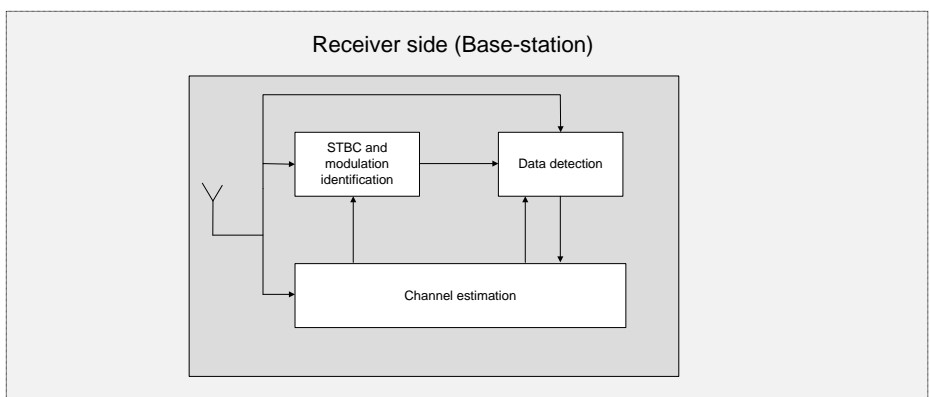

**Figure 1.** The block-diagram of the SC-FDMA system under consideration.

Each user $u$ picks a STBC type, represented by $\alpha^{(u)}$, from a set of possibilities. The time-domain vectors broadcast throughout the various time intervals, $\bar{\mathbf{c}}_{\alpha(u),\Omega(u)}^{(u,p)} = [\mathbf{c}_0^{(u,p)}, \cdots, \mathbf{c}_{N^{(u)}-1}^{(u,p)}]$, are combined to produce the signal that is sent from antenna $p$. Here $N^{(u)} = Z^{(u)} Z_1^{(u)} / Z_2^{(u)}$ and $\bar{\mathbf{c}}_{\alpha(u)}^{(u,p)}$ is linked to $\mathbf{s}^{(u)}$ via the special architecture of $\alpha^{(u)}$. We add $\alpha(u)$ and $\Omega(u)$ as subscripts to $\bar{\mathbf{c}}_{\alpha(u),\Omega(u)}^{(u,p)}$ to underline that the construction of vector $\bar{\mathbf{c}}_{\alpha(u),\Omega(u)}^{(u,p)}$ relies on STBC $\alpha(u)$ and modulation pattern $\Omega(u)$. Lastly, each user's broadcast antenna interacts with the BS over a frequency-selective channel, $\mathbf{h}^{(u,p)} = \left[ h^{(u,p)}(0), \cdots, h^{(u,p)}(L-1) \right]$, where $L$ is the channel length.

**B. Base-station**

The collected signal at the BS is a combination of the signals sent out by the transmitting users, and it is stated as

$$\mathbf{r} = \sum_{u=0}^{U-1} \sum_{p=0}^{P^{(k)}-1} \bar{\mathbf{c}}_{\alpha(u),\Omega(u)}^{(u,p)} \odot \mathbf{h}^{(u,p)} + \mathbf{w}, \tag{3}$$

where $\odot$ refers to the linear convolution between two vectors and $\mathbf{w}$ is the additive white Gaussian noise (AWGN). The objective is to exploit the collected signal $\mathbf{r}$ to figure out the form of STBC $\alpha(u)$ and modulation $\Omega(u)$ under inaccessibility of $\mathbf{h}^{(u,p)}$ for $u = 0, \cdots, U-1$, and $p = 0, \cdots, P^{(k)} - 1$. This is a necessary task in order to carry out multi-user data demodulation successfully.

**3. Proposed Identification Algorithm**

The matrix version of (3) is utilized here for the purpose of mathematical comfortability as

$$\mathbf{r} = \sum_{u=0}^{U-1} \sum_{p=0}^{P^{(k)}-1} \mathbf{C}_{\alpha(u),\Omega(u)}^{(u,p)} \mathbf{h}^{(u,p)} + \mathbf{w}, \tag{4}$$

and $\mathbf{C}_{\alpha(u),\Omega(u)}^{(u,p)}$ is an $\left( (\mathcal{N}+\nu)N^{(u)} + L - 1 \right) \times (L-1)$ matrix formed as

$$\mathbf{C}_{\alpha(u),\Omega(u)}^{(u,p)}(v_1,v_2) = \begin{cases} \bar{\mathbf{c}}_{\alpha(u),\Omega(u)}^{(u,p)}(v_1 - v_2) \\ \quad \text{for } v_1 = 0, \cdots, (\mathcal{N}+\nu)N^{(u)} + L - 1, \\ \quad \text{and } v_2 = 0, \cdots, L-1,\, v_1 \geq v_2 \\ 0 \qquad \text{otherwise,} \end{cases} \tag{5}$$

where $\mathbf{C}_{\alpha(u),\Omega(u)}^{(u,p)}(v_1, v_2)$ is the component situated at row $v_1$, and column $v_2$ of matrix $\mathbf{C}_{\alpha(u),\Omega(u)}^{(u,p)}$ and $\bar{\mathbf{c}}_{\alpha(u),\Omega(u)}^{(u,p)}(v_1 - v_2)$ is the $(v_1 - v_2)$th component of vector $\bar{\mathbf{c}}_{\alpha(u),\Omega(u)}^{(u,p)}$. Restating (4) in its condensed form yields

$$\mathbf{r} = \sum_{u=0}^{U-1} \overline{\mathbf{C}}_{\alpha(u),\Omega(u)}^{(u)} \overline{\mathbf{h}}^{(u)} + \mathbf{w}, \tag{6}$$

where $\overline{\mathbf{C}}_{\alpha(u),\Omega(u)}^{(u)} = \left[ \mathbf{C}_{\alpha(u),\Omega(u)}^{(u,0)}, \cdots, \mathbf{C}_{\alpha(u),\Omega(u)}^{(u,P^{(k)}-1)} \right]$ and $\overline{\mathbf{h}}^{(u)} = \left[ \mathbf{h}^{(u,0)\dagger}, \cdots, \mathbf{h}^{(u,P^{(k)}-1)\dagger} \right]^{\dagger}$. Here, $\dagger$ refers to the transpose action of a vector.

Optimizing the log-likelihood function with regard to the unknown parameters results in the ML strategy, as

$$\left[ \widehat{\alpha}(0), \widehat{\Omega}(0), \cdots, \widehat{\alpha}(U-1), \widehat{\Omega}(U-1) \right] = \arg \max_{\widetilde{\alpha}(0),\widetilde{\Omega}(0),\cdots,\widetilde{\alpha}(U-1),\widetilde{\Omega}(U-1)} \left\{ \log \Pr\left( \mathbf{r} | \widetilde{\alpha}(0), \widetilde{\Omega}(0), \cdots, \widetilde{\alpha}(U-1), \widetilde{\Omega}(U-1) \right) \right\}, \tag{7}$$

$u = 0, \cdots, U - 1$, $\hat{\circ}$ and $\tilde{\circ}$ are the measured and trail quantities of $\circ$, respectively, and $\Pr(\circ | \nabla)$ is the probability density function of $\circ$ given $\nabla$. The probability density function of the observations given the unknown parameters is stated as

$$\Pr\left(\mathbf{r} | \tilde{\alpha}(0), \tilde{\Omega}(0), \cdots, \tilde{\alpha}(U-1), \tilde{\Omega}(U-1)\right) = \int \cdots \int$$

$$\Pr\left(\mathbf{r} | \overline{\mathbf{C}}^{(0)}_{\tilde{\alpha}(0), \tilde{\Omega}(0)}, \overline{\mathbf{h}}^{(0)}, \cdots, \overline{\mathbf{C}}^{(U-1)}_{\tilde{\alpha}(U-1), \tilde{\Omega}(U-1)}, \overline{\mathbf{h}}^{(U-1)}\right) \times$$

$$\Pr\left(\overline{\mathbf{C}}^{(0)}_{\tilde{\alpha}(0), \tilde{\Omega}(0)}\right) \times \cdots \times \Pr\left(\overline{\mathbf{C}}^{(U-1)}_{\tilde{\alpha}(U-1), \tilde{\Omega}(U-1)}\right) \times \tag{8}$$

$$d\overline{\mathbf{C}}^{(0)}_{\tilde{\alpha}(0), \tilde{\Omega}(0)} \times \cdots \times d\overline{\mathbf{C}}^{(U-1)}_{\tilde{\alpha}(U-1), \tilde{\Omega}(U-1)} \times d\overline{\mathbf{h}}^{(0)} \times \cdots \times d\overline{\mathbf{h}}^{(U-1)}.$$

It is important to note that (8) indirectly takes an average of all potential channel paths and information symbols broadcast for arbitrary choices of $\tilde{\alpha}(u)$ and $\tilde{\Omega}(u)$, $u = 0, \cdots, U - 1$. This is because the ML recognizer is blind to the actual values of these factors. As a direct consequence, determining (8) in a real-world context is incredibly difficult. In this scenario, the SAGE process is beneficial, since it offers a smaller method to predict the ML solution in spite of the existence of nuisance parameters. This process is an improved form of the repetitive expectation-maximization mechanism, which results in a significant increase in the rate at which the algorithm converges, while maintaining the merits of efficiency and reliability [50,51]. Instead of predicting all parameters simultaneously, SAGE utilizes a set of cycles during each round. Updates to the parameter subset associated with a cycle are made by optimizing the log-likelihood of the received signal that belongs to this cycle. Thus, the multidimensional search challenge needed to optimize the likelihood function is simplified into a set of one-dimensional, computationally easy search tasks.

The following provides the theoretical specifications of the SAGE process that has been developed for computing the parameters that are taken into account. We create $U$ distinct subsets out of the unknown parameters, none of which overlap: $\left\{\alpha(0), \Omega(0), \overline{\mathbf{h}}^{(0)}\right\}$, $\cdots, \left\{\alpha(U-1), \Omega(U-1), \overline{\mathbf{h}}^{(U-1)}\right\}$. Parameter changes are applied to one user at a time. Each round consists of $U$ cycles, during which a user's subset is adjusted. The following procedures constitute the $(\zeta + 1)$th round, provided the starting estimates.

- The $u$th cycle includes updating the subset of user $u$, but nothing is changed in the subgroups of the other users.
- When the multiple access interference of each other user is deducted from the total signal received. The result is

$$\mathbf{y}_u = \mathbf{r} - \sum_{u'=0, u \neq u'}^{U-1} \mathbf{Q}^{(u')}_{\hat{\alpha}(u', \zeta), \hat{\Omega}(u', \zeta)} \hat{\overline{\mathbf{h}}}^{(u')}(\zeta), \tag{9}$$

where $\mathbf{y}_u$ is the user-$u$-collected signal portion, $\mathbf{Q}^{(u')}_{\hat{\alpha}(u', \zeta), \hat{\Omega}(u', \zeta)}$ represents the a posteriori expectation of matrix $\overline{\mathbf{C}}^{(u')}$ at the estimated values $\hat{\alpha}(u', \zeta)$ and $\hat{\Omega}(u', \zeta)$, which were computed in the previous round $\zeta$, and $\hat{\overline{\mathbf{h}}}^{(u')}(\zeta)$ is the $\zeta$th estimate of $\overline{\mathbf{h}}^{(u')}$. Notably, since the BS does not know what the information symbols are, it is unable to access matrix $\overline{\mathbf{C}}^{(u')}_{\hat{\alpha}(u', \zeta), \hat{\Omega}(u', \zeta)}$. Therefore, $\mathbf{Q}^{(u')}_{\hat{\alpha}(u', \zeta), \hat{\Omega}(u', \zeta)}$ is utilized in (9) rather than $\overline{\mathbf{C}}^{(u')}_{\hat{\alpha}(u', \zeta), \hat{\Omega}(u', \zeta)}$. We can write $\mathbf{Q}^{(u')}_{\hat{\alpha}(u', \zeta), \hat{\Omega}(u', \zeta)}$ as

$$\mathbf{Q}^{(u')}_{\hat{\alpha}(u', \zeta), \hat{\Omega}(u', \zeta)} = \mathrm{E}\left[\overline{\mathbf{C}}^{(u')}_{\hat{\alpha}(u', \zeta), \hat{\Omega}(u', \zeta)} \middle| \mathbf{y}_{u'}, \hat{\overline{\mathbf{h}}}^{(u')}(\zeta)\right], \tag{10}$$

where E[∘] is the average operator over the possible information symbols of user $u$. One expresses (10) as

$$\mathbf{Q}^{(u')}_{\widehat{\alpha}(u',\zeta),\widehat{\Omega}(u',\zeta)} = \int \overline{\mathbf{C}}^{(u')}_{\widehat{\alpha}(u',\zeta),\widehat{\Omega}(u',\zeta)} \Pr\left( \overline{\mathbf{C}}^{(u')}_{\widehat{\alpha}(u',\zeta),\widehat{\Omega}(u',\zeta)} \middle| \mathbf{y}_u, \widehat{\overline{\mathbf{h}}}^{(u')}(\zeta) \right) d\overline{\mathbf{C}}^{(u')}_{\widehat{\alpha}(u',\zeta),\widehat{\Omega}(u',\zeta)}. \tag{11}$$

- We give a description of the log-likelihood function employed to calculate the revised setting of user $u$ parameter subset as

$$\mathcal{L} = \log\Pr\left( \mathbf{y}_u \middle| \overline{\mathbf{C}}^{(u)}_{\widehat{\alpha}(u,\zeta),\widehat{\Omega}(u,\zeta)}, \widehat{\overline{\mathbf{h}}}^{(u)}(\zeta) \right). \tag{12}$$

We take into account that

$$\Pr\left( \mathbf{y}_u \middle| \overline{\mathbf{C}}^{(u)}_{\widehat{\alpha}(u,\zeta),\widehat{\Omega}(u,\zeta)}, \widehat{\overline{\mathbf{h}}}^{(u)}(\zeta) \right) \propto \exp\left( \frac{-1}{\sigma_n^2} \left\| \mathbf{y}_u - \overline{\mathbf{C}}^{(u)}_{\widehat{\alpha}(u,\zeta),\widehat{\Omega}(u,\zeta)} \widehat{\overline{\mathbf{h}}}^{(u)}(\zeta) \right\|^2 \right), \tag{13}$$

where $\sigma_n^2$ is the noise variance.

- After removing the superfluous components, we translate (12) as

$$\mathcal{L} \propto 2\Re\left\{ \mathbf{y}_u^{\ddagger} \overline{\mathbf{C}}^{(u)}_{\widehat{\alpha}(u,\zeta),\widehat{\Omega}(u,\zeta)} \widehat{\overline{\mathbf{h}}}^{(u)}(\zeta) \right\} - \widehat{\overline{\mathbf{h}}}^{(u)\ddagger}(\zeta) \overline{\mathbf{C}}^{(u)\ddagger}_{\widehat{\alpha}(u,\zeta),\widehat{\Omega}(u,\zeta)} \overline{\mathbf{C}}^{(u)}_{\widehat{\alpha}(u,\zeta),\widehat{\Omega}(u,\zeta)} \widehat{\overline{\mathbf{h}}}^{(u)}(\zeta), \tag{14}$$

where the superscript ‡ denotes the conjugate transpose of a vector and $\Re(\cdot)$ is the real part of a complex number.

- The SAGE algorithm's expectation phase employs the current estimates to derive the average value of $\mathcal{L}$ with respect to information symbols, as demonstrated in (15):

$$\mathbb{T}\left( \alpha(u), \Omega(u), \overline{\mathbf{h}}^{(u)} \middle| \widehat{\alpha}(u,\zeta), \widehat{\Omega}(u,\zeta), \widehat{\overline{\mathbf{h}}}^{(u)}(\zeta) \right) \propto$$

$$2\Re\left\{ \mathbf{y}_u^{\ddagger} \mathbf{Q}^{(u)}_{\widehat{\alpha}(u,\zeta),\widehat{\Omega}(u,\zeta)} \widehat{\overline{\mathbf{h}}}^{(u)}(\zeta) \right\} - \widehat{\overline{\mathbf{h}}}^{(u)\ddagger}(\zeta) \mathbf{Q}^{(u)\ddagger}_{\widehat{\alpha}(u,\zeta),\widehat{\Omega}(u,\zeta)} \mathbf{Q}^{(u)}_{\widehat{\alpha}(u,\zeta),\widehat{\Omega}(u,\zeta)} \widehat{\overline{\mathbf{h}}}^{(u)}(\zeta). \tag{15}$$

- The parameters of user $u$ are modified during the maximization step of the SAGE algorithm in the following fashion:

$$\left[ \widehat{\alpha}(u,\zeta+1), \widehat{\Omega}(u,\zeta+1), \widehat{\overline{\mathbf{h}}}^{(u)}(\zeta+1) \right] = \underset{\alpha(u),\Omega(u),\overline{\mathbf{h}}^{(u)}}{\arg\max}$$

$$\mathbb{T}\left( \alpha(u), \Omega(u), \overline{\mathbf{h}}^{(u)} \middle| \widehat{\alpha}(u,\zeta), \widehat{\Omega}(u,\zeta), \widehat{\overline{\mathbf{h}}}^{(u)}(\zeta) \right). \tag{16}$$

Here, we illustrate how to partition the coupled problem represented by (16) into simpler one-dimensional search tasks. Specifically, for every single couple of $(\alpha(u), \Omega(u))$, the revised value of $\overline{\mathbf{h}}^{(u)}$ is obtained by placing the function's gradient in (15) to zero a

$$\widehat{\overline{\mathbf{h}}}^{(u)}(\alpha(u), \Omega(u), \zeta+1) = \left( \mathbf{Q}^{(u)\ddagger}_{\alpha(u),\Omega(u)} \mathbf{Q}^{(u)}_{\alpha(u),\Omega(u)} \right)^{-1} \mathbf{Q}^{(u)}_{\alpha(u),\Omega(u)} \mathbf{y}_u. \tag{17}$$

- After incorporating (17) into (15), the revised values of $\alpha(u)$ and $\Omega(u)$ are recalculated as

$$\left[ \widehat{\alpha}(u,\zeta+1), \widehat{\Omega}(u,\zeta+1) \right] = \underset{\alpha(u),\Omega(u)}{\arg\max} 2\Re\left\{ \mathbf{y}_u^{\ddagger} \mathbf{Q}^{(u)}_{\alpha(u),\Omega(u)} \widehat{\overline{\mathbf{h}}}^{(u)}(\alpha(u), \Omega(u), \zeta+1) \right\}$$

$$-\widehat{\overline{\mathbf{h}}}^{(u)\ddagger}(\alpha(u), \Omega(u), \zeta+1) \mathbf{Q}^{(u)\ddagger}_{\alpha(u),\Omega(u)} \mathbf{Q}^{(u)}_{\alpha(u),\Omega(u)} \widehat{\overline{\mathbf{h}}}^{(u)}(\alpha(u), \Omega(u), \zeta+1). \tag{18}$$

The final estimate the channel vector is provided as

$$\widehat{\overline{\mathbf{h}}}^{(u)}(\zeta+1) = \widehat{\overline{\mathbf{h}}}^{(u)}\Big(\widehat{\alpha}(u,\zeta+1),\widehat{\Omega}(u,\zeta+1),\zeta+1\Big). \tag{19}$$

## 4. Practical Explanations and Implications

The following are some real-world issues that could arise from implementing the proposed iterative identification and estimation framework.

### 4.1. Expectation of Broadcast Matrices

As indicated by (10), (18), and (19), the suggested architecture depends on knowing the expectation of each user's matrix $\mathbf{Q}^{(u)}_{\alpha(u),\Omega(u)}$. A crucial question that arises is one regarding the practical computation of this matrix. In accordance with (10), it is possible to calculate $\mathbf{Q}^{(u)}_{\alpha(u),\Omega(u)}$ simply by replacing every matrix entry with its a posteriori expectation. When considering (1) and (2), the a posteriori expectation of the sent sample $s_z^{(u)}(m)$ is written as

$$\mathrm{E}\left[s_z^{(u)}(m)\bigg|\mathbf{y}_u,\widehat{\alpha}(u,\zeta),\widehat{\Omega}(u,\zeta),\widehat{\overline{\mathbf{h}}}^{(u)}(\zeta)\right] =$$

$$\sum_{l=0}^{\mathcal{N}-1}\mathrm{E}\left[f_z^{(u)}(l)\bigg|\mathbf{y}_u,\widehat{\alpha}(u,\zeta),\widehat{\Omega}(u,\zeta),\widehat{\overline{\mathbf{h}}}^{(u)}(\zeta)\right]\exp(j2\pi ml/\mathcal{N}), \tag{20}$$

where

$$\mathrm{E}\left[f_z^{(u)}(l)\bigg|\mathbf{y}_u,\widehat{\alpha}(u,\zeta),\widehat{\Omega}(u,\zeta),\widehat{\overline{\mathbf{h}}}^{(u)}(\zeta)\right] = \sum_{m=0}^{\langle\Phi(u)\rangle-1}$$

$$\mathrm{E}\left[d_z^{(u)}(m)\bigg|\mathbf{y}_u,\widehat{\alpha}(u,\zeta),\widehat{\Omega}(u,\zeta),\widehat{\overline{\mathbf{h}}}^{(u)}(\zeta)\right]\exp(-j2\pi ml/\langle\Phi(u)\rangle), \tag{21}$$

and

$$\mathrm{E}\left[d_z^{(u)}(m)\bigg|\mathbf{y}_u,\widehat{\alpha}(u,\zeta),\widehat{\Omega}(u,\zeta),\widehat{\overline{\mathbf{h}}}^{(u)}(\zeta)\right] = \sum_{\tilde{d}_z^{(u)}(m)\in\Omega(u)}$$

$$\tilde{d}_z^{(u)}(m)\Pr\left(\tilde{d}_z^{(u)}(m)\bigg|\mathbf{y}_u,\widehat{\alpha}(u,\zeta),\widehat{\Omega}(u,\zeta),\widehat{\overline{\mathbf{h}}}^{(u)}(\zeta)\right). \tag{22}$$

As (22) states, the central concern is to evaluate the a posteriori probability of $\Pr(\tilde{d}_z^{(u)}(m)|\mathbf{y}_u,\widehat{\alpha}(u,\zeta),\widehat{\Omega}(u,\zeta),\widehat{\overline{\mathbf{h}}}^{(u)}(\zeta))$. Thankfully, the decoders of contemporary error-correcting codes are able to determine this probability while they are operating in an iterative manner [47,52]. We take advantage of this likelihood to aid the suggested identification and estimation approach without adding unnecessary complexity to the decoding procedure. Figure 2 depicts the architectural block diagram of the intended approach.

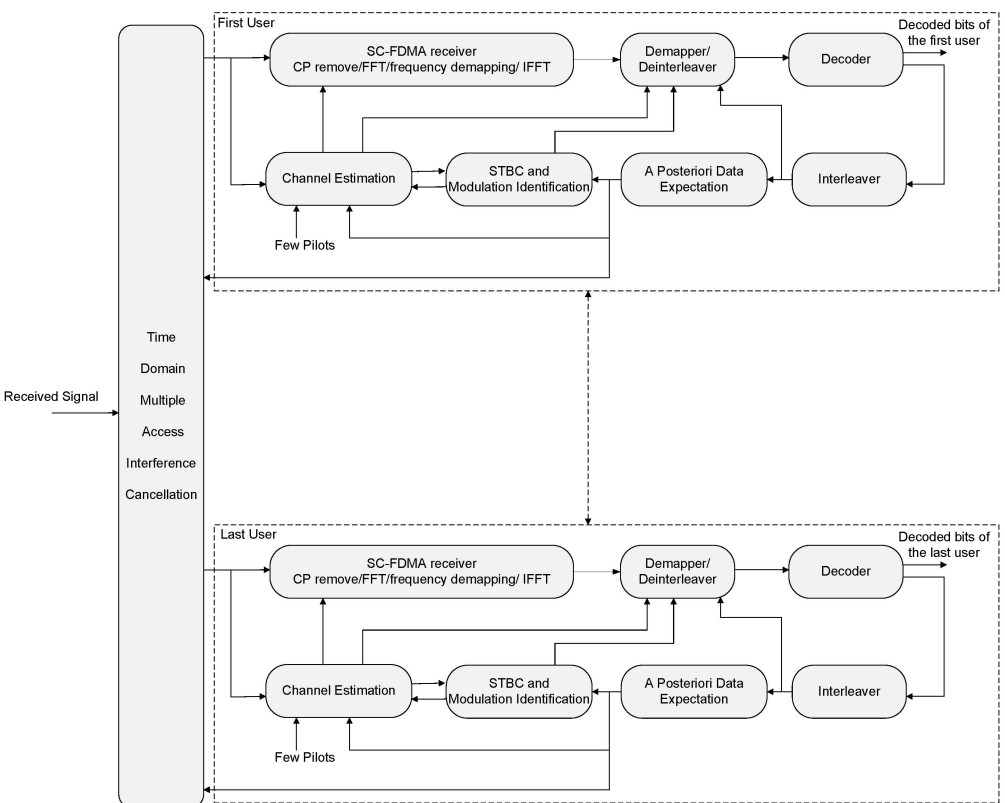

**Figure 2.** The proposed structure of the BS.

### 4.2. Channel Decoder Advancement

Every time we make adjustments to $\widehat{\alpha}(u, \zeta)$, $\widehat{\Omega}(u, \zeta)$, and $\widehat{\overline{\mathbf{h}}}^{(u)}(\zeta)$ for a particular user $u$, we have to recalculate the a posteriori probability $\Pr\left(\tilde{d}_z^{(u)}(m)\middle|\mathbf{y}_u, \widehat{\alpha}(u, \zeta), \widehat{\Omega}(u, \zeta), \widehat{\overline{\mathbf{h}}}^{(u)}(\zeta)\right)$. This calls for a reset of the channel decoder, which in turn leads to several rounds. In order to lessen this burden, we use an embedded assessment method [53], in which the channel decoder is not refreshed when the parameters $\widehat{\alpha}(u, \zeta)$, $\widehat{\Omega}(u, \zeta)$, and $\widehat{\overline{\mathbf{h}}}^{(u)}(\zeta)$ are upgraded, but the extrinsic and a priori probabilities from the previous round of the channel decoder are preserved. In this case, the extra work required by the recommended iterative process is manageable.

### 4.3. Early Estimations

Users initiate the intended SAGE-based process by sending a small number of training symbols to the BS. The first predictions $\widehat{\alpha}(u, 0)$, $\widehat{\Omega}(u, 0)$, and $\widehat{\overline{\mathbf{h}}}^{(u)}(0)$ get more accurate as the number of training symbols increases. In contrast, boosting the number of training symbols decreases the energy accessible for data symbols and raises the required bandwidth. Therefore, the ratio of training symbols to information symbols should be as little as practicable. The suggested methodology makes use of the soft information included within the soft information symbols that are delivered by the channel decoder. This allows us to produce excellent identification and estimation accuracy while having a minimum loss in throughput. The early estimations of the parameters under consideration are derived from (18) and (19) by limiting the items in $\mathbf{Q}_{\alpha(u), \Omega(u)}^{(u)}$ to the influence made by training symbols only. The proposed approach is summarized in Algorithm 1.

---

**Algorithm 1:** Summary of the proposed algorithm

---

For each user $u$:

- Use a few training symbols to construct the matrix $\mathbf{Q}^{(u)}_{\alpha(u),\Omega(u)}$ as shown in (5) for each possible combination of STBC format $\alpha(u)$ and modulation type $\Omega(u)$. Here, we set the unknown data symbols to zeros.
- Use the proposed time-domain multiple access interference cancellation algorithm, as indicated in (9).

For $\zeta = 1 : \mathcal{I}$, where $\mathcal{I}$ is the number of iterations:

1. Use (17) to determine the channel estimate under each possible pair of $\alpha(u)$ and $\Omega(u)$.
2. Use (18) to determine the estimates of $\alpha(u)$ and $\Omega(u)$.
3. Use (19) to determine the final channel estimate.
4. Perform soft data detection under the estimates of channel parameters, modulation type, and STBC format.
5. Use the soft decoder outputs to compute the a posterior expectations of the transmitted data symbols as reported in (22).
6. Use the a posteriori expectation of the transmitted data symbols to construct $\mathbf{Q}^{(u)}_{\alpha(u),\Omega(u)}$, as shown in (5). Here, we set the unknown data symbols to their a posteriori expectations.
7. Update the estimation of channel parameters, modulation type, and STBC format, as shown in (17), (18), and (19).

End (iterations)

- Perform a hard-decision criterion to recover the original transmitted data symbols.

End (users)

---

## 5. Simulation Results

The proposed STBC identification method was examined through the use of Monte Carlo simulations. The following settings were considered except where otherwise noted.

- The number of users was $U = 4$.
- The number of total subcarriers was $\mathcal{N} = 2048$.
- There were $\langle \Phi(u) \rangle = 512$, $u = 0, 1, 2, 3$ subcarriers given for every user.
- There were $\nu = 11$ samples of cyclic prefix.
- The interleaved sub-carrier method was employed.
- The assigned modulation constellation $\Omega(u)$ for each user was picked at random from a collection of 4-QAM, 16-QAM, 64-QAM, 128-QAM, 256-QAM, and 512-QAM.
- Every user employs a convolutional code of rate 0.5.
- Training symbols of size $\chi = 60$ were introduced to launch the identifying task.
- Each link, $\mathbf{h}^{(k,p)}$, between antenna $p$ of user $u$ and the BS, had 10 taps with a delay profile as indicated in [39,54].
- Every user was randomly given a code from {S1, S2, S3, S4, and S5} with transmission matrices being as follows [33,55].

$$S1(d_0, d_1) = [d_0, d_1]^\dagger, \tag{23}$$

$$S2(d_0, d_1) = \begin{bmatrix} d_0 & d_1 \\ -d_1^* & d_0^* \end{bmatrix}^\dagger, \tag{24}$$

$$S3(d_0, d_1, d_2) = \begin{bmatrix} d_0 & d_1 & d_2 \\ -d_1^* & d_0^* & 0 \\ d_2^* & 0 & -d_0^* \\ 0 & -d_2^* & d_1^* \end{bmatrix}^\dagger, \tag{25}$$

$$
S4(d_0, d_1, d_2) = \begin{bmatrix} d_0 & d_1 & \frac{d_2}{\sqrt{2}} \\ -d_1^* & d_0^* & \frac{d_2}{\sqrt{2}} \\ \frac{d_2^*}{\sqrt{2}} & \frac{d_2^*}{\sqrt{2}} & \frac{-d_0 - d_0^* + d_1 - d_1^*}{2} \\ \frac{d_2^*}{\sqrt{2}} & -\frac{d_2^*}{\sqrt{2}} & \frac{d_1 + d_1^* + d_0 - d_0^*}{2} \end{bmatrix}^{\dagger}, \tag{26}
$$

$$
S5(d_0, d_1, d_2, d_3) = \begin{bmatrix} d_0 & d_1 & d_2 & d_3 \\ -d_1 & d_0 & -d_3 & d_2 \\ -d_2 & d_3 & d_0 & -d_1 \\ -d_3 & -d_2 & d_1 & d_0 \\ d_0^* & d_1^* & d_2^* & d_3^* \\ -d_1^* & d_0^* & -d_3^* & d_2^* \\ -d_2^* & d_3^* & d_0^* & -d_1^* \\ -d_3^* & -d_2^* & d_1^* & d_0^* \end{bmatrix}^{\dagger}. \tag{27}
$$

- The probability of improper classification $P_i$ served as a merit figure for the proposed classifier and the mean-square error (MSE) was employed to evaluate the performance of channel prediction.

Figure 3 provides a visual representation of the proposed algorithm's joint modulation and STBC recognition performance over a wide range of signal-to-noise ratios (SNRs) and rounds. Additionally, Figure 4 indicates the MSE performance of the offered multiuser channel estimator. According to the results, the performance improves as the number of rounds that are utilized throughout the procedure increases; however, beyond round five, there is no visible gain made. It should be noted that an unsatisfactory performance is obtained when only a small number of training symbols are used in the initial round. When the channel decoder's output is incorporated into the identification procedure, the performance strengthens tremendously. This is due to the fact that the outputs of the decoder are fed into the proposed identifier in the same manner as if they were training symbols. The performances of the methods described in [43,44] are also displayed in Figure 3 for the sake of comparison. It has been seen that the proposed procedure is massively superior to the alternatives. This is due to the fact that the proposed strategy is optimized for use in multiuser transmissions, whereas the methods of [43,44] are intended for use in single-user transmissions.

Figure 5 depicts the performance of $P_i$ in three different scenarios in six rounds. First, we take into consideration the possibility of conducting multiuser channel predictions, as offered in (19). Second, we pay attention to the application of perfect multiuser channel predictions. The last one uses full knowledge of transmit data symbols and faultless channel estimates into account; this sets the bar for how well the proposed identifying system operates. The results demonstrate that there are no major differences between the three scenarios, especially at intermediate and high values of SNR. This is further evidence that the intended design is successful.

Figure 6 illustrates the probability of improper classification of the proposed approach for various user numbers at SNR = 14 dB and six rounds. It appears that as the number of users grows, the performance of $P_i$ suffers. This is due to a rise in residual multiple access interference, which, in turn, leads to a hindering of the operation of the suggested identifier.

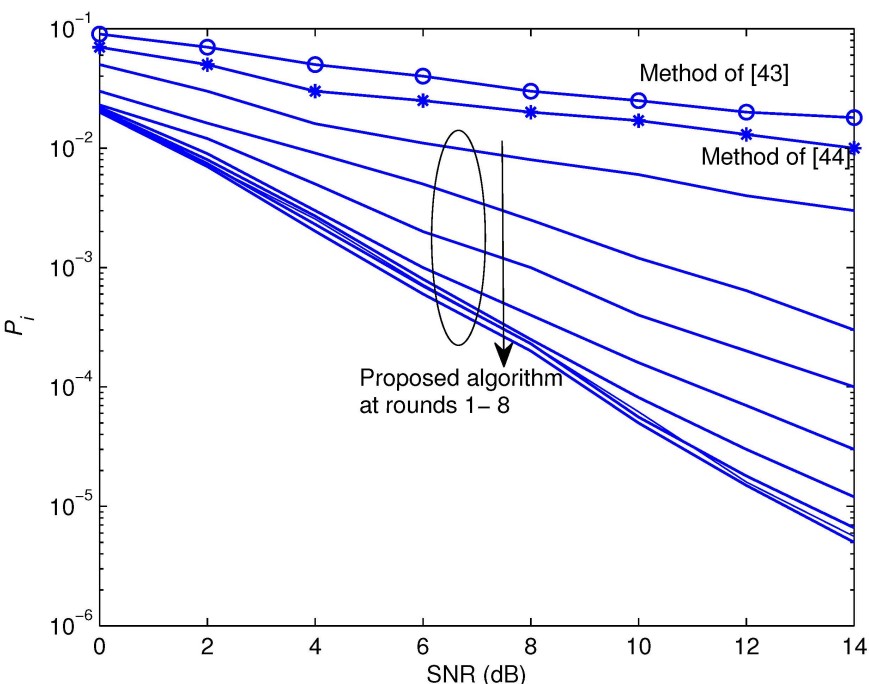

**Figure 3.** The probability of improper classification $P_i$ of the proposed classifier in comparison to the works of [43,44]. The system's parameters are $U = 4$, $\mathcal{N} = 2048$, $\langle \Phi(u) \rangle = 512$, $u = 0, 1, 2, 3$, $v = 11$, $\chi = 60$; 10 channel paths; convolutional codes of rate 0.5; interleaved sub-carrier mapping; STBC set of S1, S2, S3, S4, and S5; and a modulation set of 4-QAM, 16-QAM, 64-QAM, 128-QAM, 256-QAM, and 512-QAM.

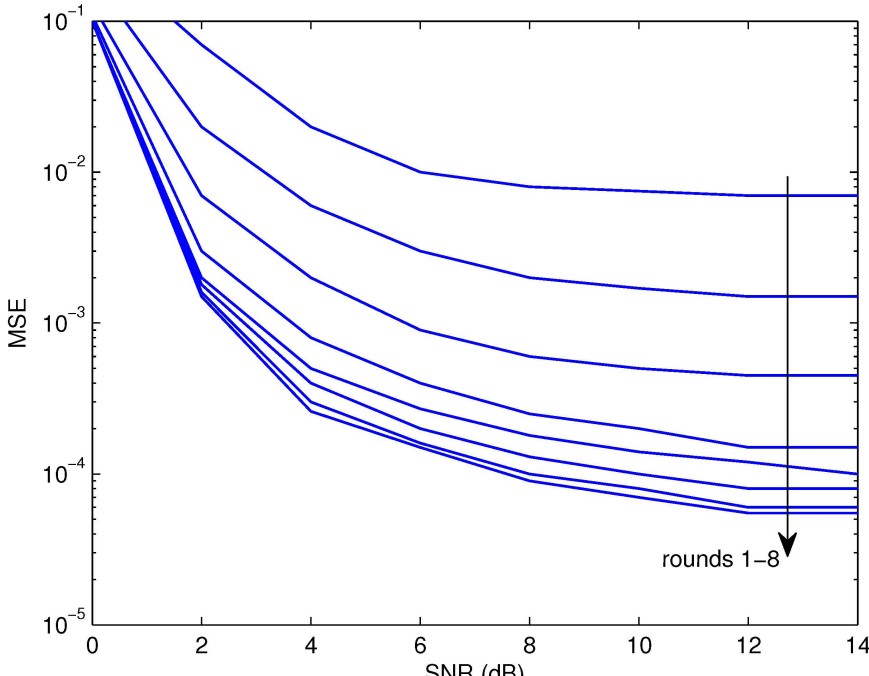

**Figure 4.** MSE performance of the proposed multiuser channel predictions. The system parameters are $U = 4$, $\mathcal{N} = 2048$, $\langle \Phi(u) \rangle = 512$, $u = 0, 1, 2, 3$, $v = 11$, $\chi = 60$, 10 channel paths, convolutional codes of rate 0.5, interleaved sub-carrier mapping, STBC set of S1, S2, S3, S4, S5, and modulation set of 4-QAM, 16-QAM, 64-QAM, 128-QAM, 256-QAM, and 512-QAM.

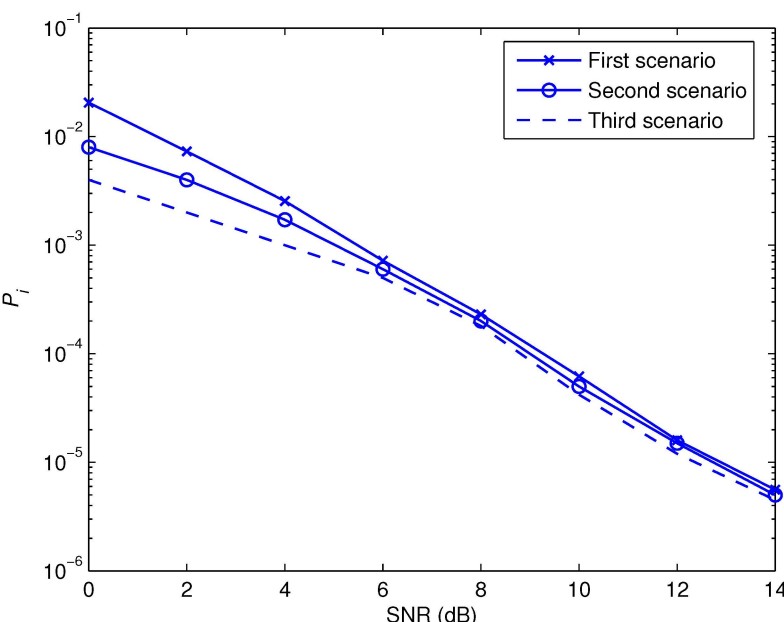

**Figure 5.** The identification performance at three scenarios. The system's parameters are SNR = 14 dB, $\mathcal{N} = 2048$, $\langle \Phi(u) \rangle = 512$, $u = 0, 1, 2, 3$, $\nu = 11$, $\chi = 60$, 10 channel paths, convolutional codes of rate 0.5, interleaved sub-carrier mapping, STBC set of S1, S2, S3, S4, S5, and modulation set of 4-QAM, 16-QAM, 64-QAM, 128-QAM, 256-QAM, and 512-QAM.

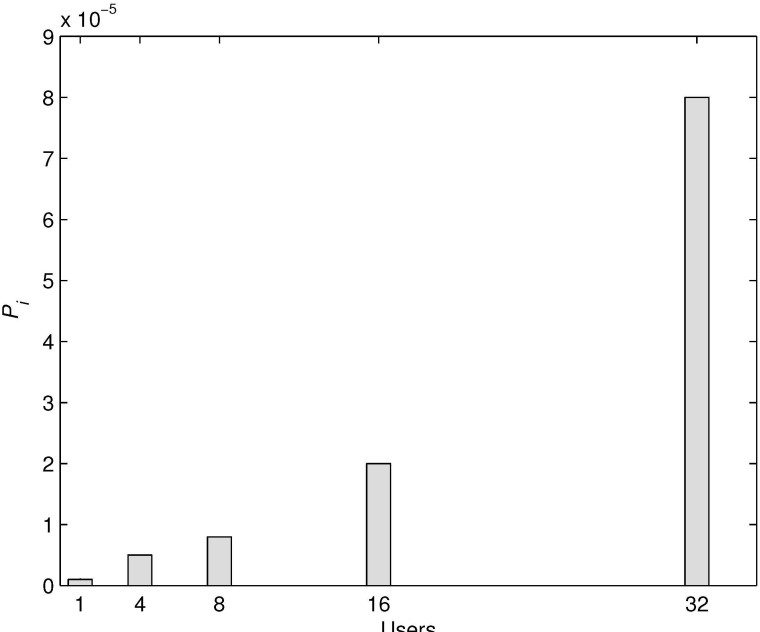

**Figure 6.** The impact of increasing the number of users on the proposed identifier. The system's parameters are $U = 4$, $\mathcal{N} = 2048$, $\langle \Phi(u) \rangle = 512$, $u = 0, 1, 2, 3$, $\nu = 11$, $\chi = 60$, 10 channel paths; convolutional codes of rate 0.5; interleaved sub-carrier mapping; an STBC set of S1, S2, S3, S4, S5; and a modulation set of 4-QAM, 16-QAM, 64-QAM, 128-QAM, 256-QAM, and 512-QAM.

It is well known that effective initiation is necessary for SAGE-based algorithms to function properly. Figure 7 illustrates the identification performance of the proposed approach as a function of the number of training symbols, $\chi$, at SNR = 6 and 14 dB. As can be observed, the intended identifier needs 60 training symbols to converge. It is important to note that the minimal level of the training symbols needed to get reliable preliminary predictions may vary based on the specifics of the system and the links. In

reality, the developer can use the suggested identification algorithm with various system and link specifications to find the minimum number of the training symbols that fulfills the specifications for each set of parameters. The obtained information is then placed in lookup tables for usage in the actual hardware. The developer also has the option of selecting a large (but still manageable) set of the training symbols that work well over a broad variety of system and connection settings at random.

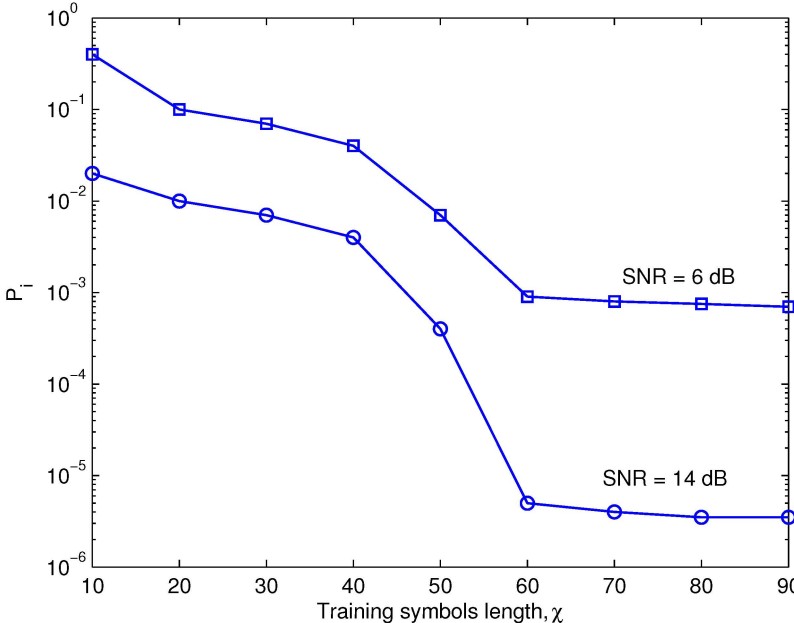

**Figure 7.** The impact of the number of the training symbols on the proposed identification algorithm. The system's parameters are $U = 4$, $\mathcal{N} = 2048$, $\langle \Phi(u) \rangle = 512$, $u = 0, 1, 2, 3$, $v = 11$, $\chi = 60$, 10 channel paths; convolutional codes of rate 0.5; interleaved sub-carrier mapping; an STBC set of S1, S2, S3, S4, and S5; and a modulation set of 4-QAM, 16-QAM, 64-QAM, 128-QAM, 256-QAM, and 512-QAM.

## 6. Conclusions

This study explored the challenges of concurrent space-time block coding (STBC) and modulation identification for multi-user upstream broadcasts in single-carrier frequency division multiple access (SC-FDMA) systems. The theoretical study exhibited that a space-alternating expectation-maximization (SAGE) strategy was utilized to put into practice the maximum-likelihood (ML) option of the parameters under discussion. The estimation of multi-user channels was undertaken as a secondary operation. The deliverable offered by the channel decoder were utilized in a sequential model so that the accuracy of identifying STBC and modulation kinds and estimating multi-user channels could be improved. The simulation results that were achieved under a variety of different scenarios, providing evidence that the offered architecture is a viable option for implementation in practical applications.

**Author Contributions:** Conceptualization, M.M. and H.M.; methodology, M.M. and H.M.; software, M.M. and H.M.; validation, M.M. and H.M.; formal analysis, M.M. and H.M.; investigation, M.M. and H.M.; resources, M.M. and H.M.; data creation, M.M. and H.M.; writing—original draft preparation, M.M. and H.M.; writing—review and editing, M.M. and H.M.; visualization, M.M. and H.M.; supervision, M.M. and H.M.; project administration, M.M. and H.M.; funding acquisition, M.M. and H.M. All authors have read and agreed to the published version of the manuscript.

**Funding:** Princess Nourah bint Abdulrahman University Researchers Supporting Project number PNURSP2023R137, Princess Nourah bint Abdulrahman University, Riyadh, Saudi Arabia.

**Institutional Review Board Statement:** Not applicable.

**Informed Consent Statement:** Not applicable.

**Data Availability Statement:** Not applicable.

**Acknowledgments:** The authors would like to acknowledge the support of Prince Sultan University for paying the Article Processing Charges (APC) of this publication.

**Conflicts of Interest:** The authors declare no conflict of interest.

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
