# Peer review of "A Powerful Joint Modulation and STBC Identification Algorithm for Multiuser Uplink SC-FDMA Transmissions"

_applsci, doi:10.3390/app13031853_

Round 1
Reviewer 1 Report
Dear Author, Congratulations for your work.
here are few suggestions,
result explanation can be further improved.
more Recent References can be added
figure 3 and 5 needs attention, parameters are not clear
Author Response
We are grateful for the constructive feedback from the Reviewer, which has helped us strengthen the manuscript. Please the response in the attached file.

Reviewer 2 Report
This study investigates the challenges of concurrent space-time block coding and modulation identification for multi-user upstream broadcasts in SC-FDMA systems, and presents a joint modulation and STBC identification algorithm for multiuser uplink SC-FDMA transmissions. Several interesting results are presented in this paper. However, I have some suggestions.
1. This paper lists six contributions. But some listed contributions are not outstanding. For example, the sixth contribution is only a feature of the proposed algorithm.
2. The second section is not well organized.
3. It is difficult to understand proposed identification algorithm. The authors should present pseudocode-based description.
Author Response
We are grateful for the constructive feedback from the Reviewer, which has helped us strengthen the manuscript. Please see the attached file.

Round 2
Reviewer 2 Report
I have no further comment.